# Angiotensin II and AT_1a_ Receptors in the Proximal Tubules of the Kidney: New Roles in Blood Pressure Control and Hypertension

**DOI:** 10.3390/ijms23052402

**Published:** 2022-02-22

**Authors:** Ana Paula de Oliveira Leite, Xiao C. Li, Sarah M. Nwia, Rumana Hassan, Jia L. Zhuo

**Affiliations:** 1Tulane Hypertension and Renal Center of Excellence, 1430 Tulane Avenue, New Orleans, LA 70112, USA; adeoliveiraleite@tulane.edu (A.P.d.O.L.); xli68@tulane.edu (X.C.L.); snwia@tulane.edu (S.M.N.); rhassan1@tulane.edu (R.H.); 2Department of Physiology, Tulane University School of Medicine, New Orleans, LA 70112, USA

**Keywords:** angiotensin II, AT_1_ receptor, hypertension, proximal tubule, sex differences

## Abstract

Contrary to public perception, hypertension remains one of the most important public health problems in the United States, affecting 46% of adults with increased risk for heart attack, stroke, and kidney diseases. The mechanisms underlying poorly controlled hypertension remain incompletely understood. Recent development in the *Cre/LoxP* approach to study gain or loss of function of a particular gene has significantly helped advance our new insights into the role of proximal tubule angiotensin II (Ang II) and its AT_1_ (AT_1a_) receptors in basal blood pressure control and the development of Ang II-induced hypertension. This novel approach has provided us and others with an important tool to generate novel mouse models with proximal tubule-specific loss (deletion) or gain of the function (overexpression). The objective of this invited review article is to review and discuss recent findings using novel genetically modifying proximal tubule-specific mouse models. These new studies have consistently demonstrated that deletion of AT_1_ (AT_1a_) receptors or its direct downstream target Na^+^/H^+^ exchanger 3 (NHE3) selectively in the proximal tubules of the kidney lowers basal blood pressure, increases the pressure-natriuresis response, and induces natriuretic responses, whereas overexpression of an intracellular Ang II fusion protein or AT_1_ (AT_1a_) receptors selectively in the proximal tubules increases proximal tubule Na^+^ reabsorption, impairs the pressure-natriuresis response, and elevates blood pressure. Furthermore, the development of Ang II-induced hypertension by systemic Ang II infusion or by proximal tubule-specific overexpression of an intracellular Ang II fusion protein was attenuated in mutant mice with proximal tubule-specific deletion of AT_1_ (AT_1a_) receptors or NHE3. Thus, these recent studies provide evidence for and new insights into the important roles of intratubular Ang II via AT_1_ (AT_1a_) receptors and NHE3 in the proximal tubules in maintaining basal blood pressure homeostasis and the development of Ang II-induced hypertension.

## 1. Introduction

Hypertension affects more than 46% of adults in the United States, significantly increasing their risk for cardiovascular diseases, stroke, and kidney failure [1,2,3]. Several classes of drugs are currently available to treat hypertension, including angiotensin-converting enzyme (ACE) inhibitors, angiotensin II (Ang II) receptor blockers (ARBs), calcium channel inhibitors, β-blockers, and loop diuretics. These antihypertensive drugs act to either inhibit the renin–angiotensin system (RAS), cause blood vessel vasodilatation, induce natriuresis and diuresis, or suppress sympathetic nerve activity [4,5,6,7]. Despite widespread treatments with these antihypertensive drugs, only approximately 50% of patients have attained adequate blood pressure control, while the rest continue to develop apparent treatment-resistant hypertension (aTRH) [6]. aTRH has been defined as uncontrolled blood pressure while taking three classes of antihypertensive drugs or taking four classes of antihypertensive drugs regardless of blood pressure control [4,5,7]. With nearly 20% of American adults affected by aTRH, the mechanisms of hypertension, whether controlled or poorly controlled, remain incompletely understood and warrants further studies [4].

The pressure-natriuresis response is one of the major mechanisms by which the kidneys control blood pressure and body salt and fluid homeostasis in response to the changes in renal arterial pressure [3,4,8]. A significant increase in arterial blood pressure is expected to trigger the pressure-natriuresis response that, in turn, alters interstitial hydrostatic pressure, proximal tubule Na^+^ transport, and renal medullary blood flow to induce diuresis and natriuresis responses. This is followed by blood pressure returning to control. Conversely, when blood pressure falls significantly, the pressure-natriuresis response is suppressed, antidiuretic and anti-natriuretic responses are augmented, and proximal tubule Na^+^ reabsorption is increased [3,4,8]. These responses work together to restore blood pressure back to control. However, how the pressure-natriuresis response is regulated and its potential mediators remain poorly understood. Recent studies suggest that the intratubular RAS, especially Ang II via activation of AT_1_ (AT_1a_) receptors in the proximal tubules of the kidney, plays a key role in basal blood pressure control and the development of Ang II-induced hypertension by regulating proximal tubule Na^+^ reabsorption and the pressure-natriuresis response [9,10,11].

Recently, the XIII International Symposium on Vasoactive Peptides was held on 15–17 October 2021, which celebrated recent advances in studying the roles of vasoactive peptides, especially the renin–angiotensin system, in cardiovascular, neural, kidney, and blood pressure control, inflammation and the COVID-19 pandemic. Based on a lecture given at this symposium, the objective of this article is to briefly review and discuss recent studies using unique mouse models with proximal tubule-specific deletion (knockout) of Ang II type 1a receptors (AT_1a_) or its major downstream target protein, the Na^+^/H^+^ exchanger 3 (NHE3) in the kidney [12,13,14,15,16]. These studies have revealed important roles of AT_1a_ receptors, acting alone or via NHE3, in the proximal tubules in maintaining basal blood pressure homeostasis, the pressure-natriuresis response, and the development of Ang II-induced hypertension. This new knowledge improves our understanding of the renal mechanisms of blood pressure regulation as well as Ang II-induced hypertension and suggests that both AT_1a_ receptors and NHE3 may be potential therapeutic targets in treating hypertension in humans.

## 2. The Proximal Tubules Are Major Tubular Segments Expressing a Robust Renin–Angiotensin System in the Kidney

It has long been recognized that the proximal tubules express all major components of the RAS including key enzymes (i.e., renin and ACE) and sole substrate (i.e., angiotensinogen) that are required to generate Ang II and the receptors (i.e., AT_1_ and AT_2_) that mediate the actions of Ang II [17,18,19,20]. The proximal tubules also express abundant enzyme (ACE2) and aminopeptidases (i.e., APA and APN) that metabolize Ang II or its downstream active fragments, Ang III or Ang (3–8) [21,22,23,24]. Ang II is the most potent effector of all angiotensin peptides and plays the most critical roles in the kidney to regulate blood flow, glomerular filtration, and tubular transport [13,25,26]. Most of the well-recognized effects of Ang II in the kidney are mediated by AT_1_ receptors [11,13,15,25,26,27,28,29], while AT_2_ receptors play a relatively moderate role in mediating the natriuretic response in the proximal tubules [16,30,31,32,33]. The molecular cloning of AT_1_ and AT_2_ receptors, and the development of Ang II receptor blockers (ARBs) for treatment of hypertension, diabetic nephropathy, and other kidney diseases represent one of the most significant breakthroughs in the Ang II receptor research field over last 3 decades.

The AT_1_ receptor is molecularly classified into two subtypes in rodents, AT_1a_ and AT_1b_, based on their molecular structures [34,35,36,37]. However, humans express only one AT_1_ receptor corresponding to AT_1a_ receptors in rodents. The AT_1a_ receptor was first cloned from rat vascular smooth muscle cells with its cDNA encoding a 359 amino-acid protein, a molecular structure typical of seven transmembrane GPCR [36,37]. The AT_1b_ receptor was subsequently cloned from bovine adrenal cells with its cDNA sharing a 94% identical amino acid sequence of the AT_1a_ receptor [34,35]. Most AT_1_ receptors are of the AT_1a_ subtype accounting for >90% of the AT_1_ receptor family in the kidney and other tissues [25,26,28,38,39], whereas the AT_1b_ receptor accounts for approximately 5–10%, and its expression is mainly confined to the adrenal glands, the kidney, and brain [40,41,42]. Thus, it is not surprising that most of the well-recognized effects of Ang II are mediated by AT_1a_ receptors, whereas AT_1b_ receptors play a small role [13,25,26,28,38].

AT_1_ and AT_2_ receptors have been localized in the rat, mouse, and human kidney using quantitative in vitro or in vivo autoradiography (Figure 1) [43,44,45,46,47,48], in situ hybridization histochemistry [49,50,51], and immunohistochemistry, respectively [52,53,54]. The localization of AT_1_ (AT_1a_) receptors in the kidney by radioreceptor binding and autoradiography is the gold-standard approach based on the direct GPCR and ligand pharmacology principles [43,44,45,46,47,48]. This technique has revealed a distinct anatomical distribution or localization of these receptors in the kidneys of rat, mouse, rabbit, monkey, or humans [43,44,45,46,47,55]. Specifically, autoradiographs show a very high density of AT_1_ receptors in the glomerulus, corresponding to mesangial cells, endothelial cells, and podocytes [43,44,45,46]. Although not as high as in the glomerulus, a moderately high level of AT_1_ receptors is localized in the intervening outer cortex corresponding to proximal convoluted tubules [44,45,46,47]. Interestingly, a high density of AT_1_ receptors is also localized in the longitudinal bands traversing the inner stripe of the outer medulla associated with the vasa recta bundles and type 1 renomedullary interstitial cells [47,56]. By contrast, the inner cortex, the outer stripe of the outer medulla, and the entire inner medulla especially toward the tip of the inner medulla express low to undetectable levels of AT_1_ receptors [43,44,45,46,47,48]. In comparison, the expression of AT_2_ receptors is very low throughout the kidney of adult rodents or humans except in the proximal tubules and blood vessels (Figure 1) [47,48].

Other techniques have also been used to localize AT_1_ and AT_2_ receptors in the kidney. In situ hybridization histochemistry offers a high sensitivity and specificity to localize AT_1_ (AT_1a_) receptor mRNAs throughout the kidney including the blood vessels, glomerulus, proximal tubules, loop of Henle, distal tubules, and collecting ducts [49,50,51]. A recent study using the novel RNA scope technique reported similar findings throughout the kidney with AT_1_ receptor mRNAs observed in mesangial cell, juxtaglomerular cells, proximal tubule cells, interstitial cell, and late afferent and early efferent arterioles [57]. The localization of AT_1_ and AT_2_ receptor proteins in the kidney has also been studied, primarily using immunohistochemistry with AT_1_ and AT_2_ receptor antibodies [52,53,54]. The localization of AT_1_ (AT_1a_) and AT_2_ receptor mRNAs or proteins in most, if not all, kidney cells or structures by these techniques may be different from the localization of AT_1_ receptor binding sites by radioreceptor or autoradiographic binding [43,44,45,46,47,48]. Although these techniques are very sensitive to detect AT_1_ and AT_2_ receptor mRNAs or proteins in the kidney, the specificity of these approaches, especially using commercially available AT_1_ or AT_2_ receptor antibodies for immunohistochemistry, remains controversial [58,59,60].

## 3. The Proximal Tubules Are Major Tubular Segments Expressing the Most Robust NHE3 Abundance in the Kidney, a Major Downstream Target of AT_1_ (AT_1a_) Receptor Activation

In contrast to the general perception that NHE3 is expressed in every tissue in the body, only two major organs or tissues express NHE3 most abundantly, i.e., the gastrointestinal tract (gut) and the kidney. In the gut, NHE3 is primarily expressed in the small intestines and much less in the stomach and large intestines or colon [61,62]. In the kidney, NHE3 is primarily expressed in proximal convoluted and straight tubules, and to a moderate extent, in the loop of Henle and is virtually not expressed in collecting ducts [61,62,63,64,65,66]. NHE3 is primarily localized on the apical membranes of the proximal tubules and loop of Henle under physiological conditions and mediates electroneutral Na^+^ entry into the cells from the lumen and H^+^ extrusion from the cells in the proximal tubules [67,68]. After entry into proximal tubule cells, intracellular Na^+^ ions are returned to the blood via the Na^+^/K^+^-ATPase, i.e., the Na^+^ and K^+^ pump on the basolateral membranes. Low blood pressure, loss of blood, sodium depletion, or salt wasting activate intratubular RAS expression and generation of Ang II that, in turn, activates AT_1_ (AT_1a_) receptors to stimulate NHE3 expression and increase Na^+^ reabsorption in the proximal tubules. These responses, along with other central neural, cardiovascular, and humoral factors help restore body salt and fluid balance and blood pressure homeostasis. By contrast, acute and chronic increases in blood pressure induces NHE3 endocytosis or redistribution from apical membranes, which augments the pressure-natriuresis response and induces diuresis and natriuresis that helps restore blood pressure to control [13,14,15].

NHE3 in the proximal tubules is the major downstream target of intratubular Ang II via AT_1_ (AT_1a_) receptors in the proximal tubules [15,69]. Physiologically, Ang II binds and activates AT_1_ (AT_1a_) and AT_2_ receptors in the proximal tubules, with AT_1_ (AT_1a_) receptors playing a dominant role [13,15] and AT_2_ receptors playing a smaller counterregulatory role [30,70]. Previous studies have shown that activation of AT_1_ (AT_1a_) receptors by Ang II mediates G protein-coupled, PKCα, IP_3_, and Ca^2+^/calmodulin-dependent protein kinase II signaling to induce NHE3 expression and activity in cultured proximal tubule cells [55,71,72]. Other anti-natriuretic factors, such as glucocorticoids [73,74], glucagon, and insulin [75,76], also stimulate proximal tubule Na^+^ reabsorption, in part, by upregulating NHE3 expression in the kidney. Conversely, Ang II and its major metabolite Ang III reportedly activate AT_2_ receptor-mediated NO/cGMP signaling to induce NHE3 endocytosis and increased urinary Na^+^ excretion [30,70]. The natriuretic peptide dopamine has also been shown to inhibit proximal tubule Na^+^ reabsorption and induces natriuresis by inhibiting NHE3 expression in the kidney [67,77,78,79,80]. Nevertheless, marked upregulation of intratubular Ang II/AT_1_ (AT_1a_) receptors/NHE3 signaling in the proximal tubules plays a more dominant role in the development of Ang II-induced hypertension.

## 4. AT_1_ (AT_1a_) Receptors in the Proximal Tubules of the Kidney Are Required for Maintaining Basal Blood Pressure Homeostasis

It has long been recognized that the proximal tubules play a key role in overall blood pressure regulation and the development of hypertension. This recognition is based on the fact that approximately 65–70% of Na^+^ and fluid is reabsorbed by the proximal tubules alone; thus, increases or decreases in proximal tubule reabsorption will exert a significant impact on blood pressure homeostasis [20]. Several previous studies have suggested that distal tubular segments may also play important roles in blood pressure regulation and the development of hypertension due to the fact of their ability to fully compensate for any increases or decreases in Na^+^ delivery from the proximal tubules [81,82,83]. However, recent studies from our and other labs suggest otherwise, because distal nephron segments fail to compensate for the loss of AT_1_ (AT_1a_) receptors or its major downstream target Na^+^ transporter NHE3 in the proximal tubules, leading to inhibition of proximal tubule Na^+^ reabsorption, natriuretic response, and lower basal blood pressure [13,15,27].

As mentioned previously, all major components of the RAS, including angiotensinogen, renin, ACE, and AT_1_ and AT_2_ receptors, have all been localized in the proximal tubules of the kidney [17,18,19,20]. Ang II concentrations in the proximal tubules are much higher than in the circulation under both the physiological and hypertensive states [84,85,86,87]. This may be due to several factors, including, but not limited to, (a) the expression of the substrate angiotensinogen and key enzymes renin and ACE for the generation of Ang II onsite; (b) the capacity of AT_1_ (AT_1a_) receptor- and the endocytic receptor megalin-mediated accumulation of circulating and tissue paracrine Ang II by the proximal tubules [88,89,90]; and (c) the feedforward regulatory mechanisms of intratubular RAS in the proximal tubules during the development of Ang II-induced hypertension [87,91]. Thus, it is expected that Ang II in the proximal tubules not only acts physiologically to stimulate proximal tubule reabsorption of sodium and fluid, maintain body sodium and fluid balance and basal blood pressure homeostasis, but also promotes sodium retention in hypertension via the actions of AT_1_ (AT_1a_) receptors [56,92,93].

Although previous studies have shown that Ang II has biphasic effects to regulate proximal tubule sodium transport based on in vivo micropuncture experiments [93,94] or in vitro proximal tubule perfusion studies [78,95], whether these local effects alter systemic blood pressure has not been studied by these approaches. Whole body loss of function (deletion) or gain of function (overexpression) of the RAS may also be unable to determine the roles of intratubular Ang II and AT_1_ (AT_1a_) receptors in the proximal tubules of the kidney on basal blood pressure level and the development of Ang II-induced hypertension. At the whole kidney level, Crowley et al. were instrumental in demonstrating a key role of the kidney RAS in blood pressure control and in Ang II-induced hypertension using the cross-kidney transplantation approach between wild-type and whole-body *Agtr1a^-/-^* mice [10,96]. Their study elegantly showed that transplantation of *Agtr1a^-/-^* mouse kidneys into wild-type mice lowered basal blood pressure and attenuated Ang II-induced hypertension, whereas transplantation of wild-type mouse kidneys into *Agtr1a^-/-^* mice elevated blood pressure [10,96]. These studies directly support a key role for kidney AT_1_ (AT_1a_) receptors in blood pressure control and Ang II-induced hypertension, but the role of intratubular AT_1_ (AT_1a_) receptors in the proximal tubules was not determined in these studies.

Two other instrumental studies have determined the role of AT_1_ (AT_1a_) receptors in the proximal tubules of the kidney using the *Cre/LoxP* approach with two different “proximal tubule-specific” promoters [27,97]. Gurley et al. used the *PEPCK-Cre*/*Agtr1a* flox [27], whereas Li et al. used the *KAP2-iCre*/*Agtr1a* flox approach to generate proximal tubule-specific AT_1a_-knockout mutant mice [97]. Both studies showed that intratubular AT_1a_ receptors regulate blood pressure with or without compensatory expression of NHE3, sodium and phosphate cotransporter 2 (NaPi_2_), Na^+^/K^+^-ATPase, sodium chloride cotransporter (NCC), Na^+^:K^+^:Cl^2-^ cotransporter 2 (NKCC2), and epithelial sodium channel (ENaC) in proximal or distal nephron segments [27,97]. However, PEPCK has been known to express, to some extent, in tissues beyond the proximal tubules, including other segments of the nephron and epithelial cells in many other extra-renal tissues such as the liver, white and brown fat, jejunum, ileum, and sublingual gland [98,99]. Likewise, the expression of renal androgen-regulated protein, KAP2, is not only confined to the proximal tubules, but also expressed in the outer medulla of the kidney and other tissues that physiologically respond to androgen [100,101]. While these studies suggest an important role of AT_1_ (AT_1a_) receptors in the proximal tubules, the specificity of the *PEPCK* or *KAP2* promoters to the proximal tubules remain to be further determined.

Against this background, our lab has recently taken a different approach to generate a new mutant mouse model (PT-*Agtr1a^-/-^*) with proximal tubule-specific deletion of AT_1_ receptors (AT_1a_) in the kidney using the *iL1-SGLT2-Cre*/*Agtr1a* flox approach [16,28]. The scientific premise for using the *iL1-SGLT2-Cre*/*Agtr1a* flox approach was based on the findings that SGLT2, the sodium and glucose cotransporter 2, is more specific to the proximal tubules with its expression primarily, if not exclusively, in the S1 and S2 segments of the proximal tubules [13,14,97,102]. No SGLT2 is expressed in distal nephron segments beyond the end of the proximal tubules or in other tissues such as brain, blood vessels, heart, or liver [102,103]. With this new approach, our studies consistently showed that selective deletion of *Agtr1a* in the proximal tubules led to an appropriate 15 ± 3 mmHg decrease in basal systolic, diastolic, and mean arterial blood pressure in both male and female PT-*Agtr1a^-/-^* mice when compared to wild-type mice (Figure 2) [16,28]. By contrast, deletion of AT_1a_ receptors in all tissues of the body led to an approximately 30 ± 5 mmHg lower systolic, diastolic, and mean arterial blood pressure in global *Agtr1a^-/-^* mice [11,16,28,55]. No significant sex differences in this basal blood pressure phenotype were observed in either global *Agtr1a^-/-^* or proximal tubule-specific PT-*Agtr1a^-/-^* mice [12,28]. The blood pressure-lowering effect in PT-*Agtr1a^-/-^* mice was associated with significant diuretic and natriuretic responses and increases in the basal glomerular filtration rate as well as the pressure-natriuresis response due to the inhibition of Na^+^ and fluid reabsorption in the proximal tubules (Figure 2) [28,104,105]. Our data strongly support the hypothesis that intratubular Ang II via actions on AT_1a_ receptors in the proximal tubules, indeed, play a key role in maintaining basal blood pressure homeostasis, and loss of AT_1a_ receptor function in the proximal tubules would lower basal blood pressure.

## 5. AT_1_ (AT_1a_) Receptors in the Proximal Tubules of the Kidney Are Required for the Development of Ang II-Induced Hypertension

The development of Ang II-induced hypertension has been extensively studied in different animal models with AT_1_ (AT_1a_) receptors playing the fundamental role [12,27,71,97,104]. AT_1_ (AT_1a_) receptors are expressed widely, although to different levels, in the brain, adrenal glands, and cardiovascular and kidney tissues; thus, the mechanisms underlying Ang II-induced hypertension are expected to involve different tissues or pathways [16,39]. We and others have recently determined the direct contributions of intratubular Ang II via AT_1_ (AT_1a_) receptors in the proximal tubules in the development of Ang II-induced hypertension by comparing global, kidney-, and proximal tubule-specific *Agtr1a^-/-^* mice [10,11,12,16,27,28,97,104]. As discussed previously in the cross-kidney transplantation study, Crowley et al. reported that kidney AT_1_ (AT_1a_) receptors played virtually equivalent roles in AT_1_ receptor actions in the kidney and in extrarenal tissues to determining the level of blood pressure and the development of Ang II-infused hypertension [10,96]. Gurley et al. showed that deletion of AT_1a_ receptors in mice significantly attenuated Ang II-induced hypertension in their mouse model generated using the *PEPCK-Cre*/*Agtr1a* flox approach [27]. By contrast, Li H. et al. demonstrated that Ang II action via the proximal tubule AT_1a_ receptors was not a significant component of the acute pressor action of increased circulating Ang II, i.e., in Ang II-induced hypertensive response and in their mouse model generated using the *KAP2-iCre*/*Agtr1a* flox approach [97]. We recently studied the roles and mechanisms responsible for the development of Ang II-induced hypertension using global and proximal tubule-specific *Agtr1a^-/-^* mice [12,16,28]. We induced Ang II-induced hypertension in male and female wild-type, global *Agtr1a^-/-^*, and proximal tubule-specific *Agtr1a^-/-^* mice by infusing exogenous Ang II (0.5–1.5 mg/kg/day, i.p.) or by adenovirus-mediated overexpressing an intracellular cyan fluorescent protein tagged Ang II fusion protein, Ad-sglt2-ECFP/Ang II, selectively in the proximal tubules using the SGLT2 promoter [16,28]. In both scenarios, systemic Ang II infusion or proximal tubule-specific expressed Ang II fusion proteins significantly increased systolic, diastolic, and mean arterial blood pressure in wild-type mice, as expected. Global deletion of AT_1_ (AT_1a_) receptors completely prevented the development of hypertension by systemic or intratubular/intracellular Ang II, also as expected, because AT_1_ (AT_1a_) receptors mediated most if not all well-recognized hypertensive responses. By contrast, Ang II-induced hypertension by systemic Ang II infusion was markedly attenuated by ~50% but not completely blocked in proximal tubule-specific PT-*Agtr1a^-/-^* mice (Figure 3) [12,28]. Interestingly, increases in blood pressure induced by proximal tubule-specific expression of Ang II fusion proteins in wild-type mice were completed blocked in proximal tubule-specific PT-*Agtr1a^-/-^* mice [16,28].

How increases in blood pressure by proximal tubule-specific overexpression of intracellular Ang II fusion protein were completely blocked, whereas Ang II-induced hypertension by systemic Ang II infusion was attenuated by only ~50% in PT-*Agtr1a^-/-^* mice, remains incompletely understood. However, both systemic and proximal tubule-specific AT_1_ (AT_1a_) receptor-dependent mechanisms are expected to mediate the development of Ang II-induced hypertension [10,13,26,27,29]. Deletion of AT_1_ (AT_1a_) receptors selectively in the proximal tubules of PT-*Agtr1a^-/-^* mice only blocks AT_1_ (AT_1a_) receptor- and NHE3-mediated stimulation of proximal tubule Na^+^ reabsorption, but has no effects beyond the proximal tubules in the kidney and other extrarenal tissues [14,15,27,29,71]. Conversely, in the studies with proximal tubule-specific overexpression of intracellular Ang II fusion protein, intracellular Ang II does not induce systemic effects. This is because intracellular Ang II fusion protein only activates intracellular AT_1_ (AT_1a_) receptors confined to the proximal tubules to stimulate proximal tubule Na^+^ reabsorption, which elevates blood pressure [71,106,107]. Thus, deletion of AT_1_ (AT_1a_) receptors selectively in the proximal tubules of PT-*Agtr1a^-/-^* mice is expected completely to block intracellular Ang II-induced blood pressure increases by blocking intracellular Ang II-stimulated proximal tubule Na^+^ reabsorption.

## 6. Sex Differences in Ang II-Induced Hypertension

Recently, sex differences in genetics, biology, physiology, and cardiovascular, kidney, and hypertensive diseases have attracted widespread attention and extensive research focus [108,109,110,111,112,113]. This is largely due to the recent mandates in the NIH Policies for Biomedical Research to consider and include gender- and/or sex-related factors as biological variables in all experimental designs [114,115,116,117]. Although great progress has been made in sex differences in many other research fields, sex differences in basal blood pressure control or in Ang II-induced hypertension have been inconsistent between different studies or animal models. Specifically, some sex differences have been found in the blood pressure or tubuloglomerular feedback (TGF) responses to Ang II [108,109,110,111], increases in NaCl cotransporter or ENaC expression in response to Ang II [118,119], the development of Ang II-induced hypertension [113], diabetic nephropathy [120], or abdominal aortic aneurysms [121]. However, whether sex differences are involved in intratubular Ang II and its AT_1a_ receptors in the proximal tubules in the development of Ang II-induced hypertension has not been studied previously.

Against this background, we have recently tested the hypothesis that there are significant sex differences in the roles of intratubular Ang II via AT_1a_ receptors in the proximal tubules during the development of Ang II-induced hypertension [12]. Specifically, we hypothesized that AT_1a_ receptors in the proximal tubules of female mice contribute less to Ang II-induced hypertension than those in male mice. To test this hypothesis, we directly compared the blood pressure, glomerular, and tubular responses to Ang II-induced hypertension in adult male and female wild-type and mutant mice with proximal tubule-specific knockout of AT_1a_ receptors (PT-*Agtr1a^-/-^*) [12]. When adult male and female wild-type and PT-*Agtr1a^-/-^* mice were infused with or without an identical pressor dose of Ang II via osmotic pump for 2 weeks (1.5 mg/kg/day, i.p.), we found that basal systolic, diastolic, and mean arterial pressure were approximately 13 ± 3 mmHg lower, while basal 24 h urinary Na^+^ excretion was significantly higher in both male and female PT-*Agtr1a^-/-^* mice than wild-type controls without significant sex differences in the same strain (Figure 4). Both male and female wild-type mice developed marked hypertension with similar magnitudes of the pressor responses to Ang II, also without significant sex differences. Furthermore, Ang II-induced hypertension was equally attenuated in male and female PT-*Agtr1a^-/-^* mice with or without concurrent blockade of AT_1_ receptors with losartan [28]. Finally, Ang II-induced glomerular and tubulointerstitial injury was attenuated in both male and female PT-*Agtr1a^-/-^* mice. Taken together, our study shows that deletion of AT_1a_ receptors in the proximal tubules of the kidney attenuated Ang II-induced hypertension and kidney injury without revealing significant sex differences.

## 7. Roles of Proximal Tubule Atrial Natriuretic Peptide, Dopamine, or Prorenin Receptors in Ang II-Induced Hypertension

Although the basal blood pressure phenotypes and their responses to Ang II-induced hypertension in proximal tubule-specific PT-*Agtr1a^-/-^* mice are primarily due to the loss of AT_1a_ receptors in the proximal tubules, the roles or involvement of other intratubular vasoactive factors may also be considered. One of these factors is G protein-coupled receptors for atrial natriuretic factor or peptide (ANP), especially NPR_A_. ANP receptors (NPR_A_) are expressed strongly in the proximal tubules in addition to the glomerulus, distal tubules, and collecting ducts [122,123,124], and ANP acts on NPR_A_ in the proximal tubules of the kidney to counteract the effects of Ang II by inhibiting proximal tubule Na^+^ reabsorption [125,126,127]. Thus, it is expected that deletion of AT_1a_ receptors in the proximal tubules may augment the natriuretic response to NPR_A_-mediated effects in the proximal tubules. However, no such study has been reported to determine whether ANP-induced natriuretic response is potentiated in PT-*Agtr1a^-/-^* mice. The other factor is dopamine receptors in the proximal tubules. Dopamine receptors are also expressed in the proximal tubules of the kidney, which mediate the natriuretic responses to dopamine administration and counteract the anti-natriuretic effects of AT_1_ (AT_1a_) receptors [128,129,130]. Prorenin receptors (PRR) are also expressed in the proximal tubules of the kidney in addition to the glomerulus, distal tubules, and collecting ducts [131,132]. The roles of PRR in the proximal tubules remain unknown. Soluble PRR reportedly promotes the fibrotic response in renal proximal tubule epithelial cells in vitro via the Akt/β-catenin/Snail signaling pathway [131], whereas deletion of nephron PRR (ATP6AP2) has been shown to impair proximal tubule function in mice [132]. However, reviewing and discussing the roles of NPR_A_, dopamine receptors, or PRR in the proximal tubules of the kidney in details are beyond the scope of this article. Future studies are necessary to confirm whether NPR_A_, PRR, or dopamine receptors are involved in mediating basal blood pressure and natriuretic phenotypes and in the development of Ang II-induced hypertension.

## 8. Conclusions and Perspectives

In summary, we have gained new insights into the important roles of intratubular Ang II and AT_1_ (AT_1a_) receptors in the proximal tubules of the kidney in maintaining basal blood pressure homeostasis and the development of Ang II-induced hypertension. Significant progress has been made directly in a few key areas over the last three decades, but other studies may also independently contribute to the progresses. Specifically, early in vivo proximal tubule micropuncture experiments in rats [93,94] and in vitro isolated proximal tubule perfusion studies in rabbits or rats [78,95] established important milestones by demonstrating direct biphasic effects of Ang II on proximal tubule Na^+^ transport function. This was followed by subsequent loss or gain of AT_1_ (AT_1a_) receptor function studies in mice using the kidney-cross transplantation between wild-type and *Agtr1a^-/-^* mice, firmly confirming the critical role of kidney AT_1_ (AT_1a_) receptors in blood pressure control and hypertension [10,96]. However, these studies have not be able to determine the direct role of intratubular Ang II and AT_1_ (AT_1a_) receptors in the proximal tubules in blood pressure control and Ang II-induced hypertension. Recent development in the *Cre/LoxP* approach has significantly helped advance new insights into the role of proximal tubule Ang II and AT_1_ (AT_1a_) receptors. This novel approach has provided us and others with an important tool to generate novel mouse models with proximal tubule-specific loss (deletion) or gain of the function (overexpression) [12,13,14,16,27,28,97]. These new studies have consistently demonstrated that deletion of AT_1_ (AT_1a_) receptors [12,15,16] or its direct downstream target, NHE3, selectively in the proximal tubules of the kidney [13,14] lowers basal blood pressure, increases the pressure-natriuresis response, and induces natriuretic responses, whereas overexpression of an intracellular Ang II fusion protein or AT_1_ (AT_1a_) receptors selectively in the proximal tubules increases proximal tubule Na^+^ reabsorption, impairs the pressure-natriuresis response, and elevates blood pressure [71,106,107]. Furthermore, since the development of Ang II-induced hypertension by systemic Ang II infusion or by proximal tubule-specific overexpression of an intracellular Ang II fusion protein was attenuated in mutant mice with proximal tubule-specific deletion of AT_1a_ [12,16,28] or NHE3 [13,69], we may conclude that intratubular Ang II via AT_1a_ and NHE3 in the proximal tubules play a key role in maintaining basal blood pressure and the development of Ang II-induced hypertension.

## Figures and Tables

**Figure 1 ijms-23-02402-f001:**
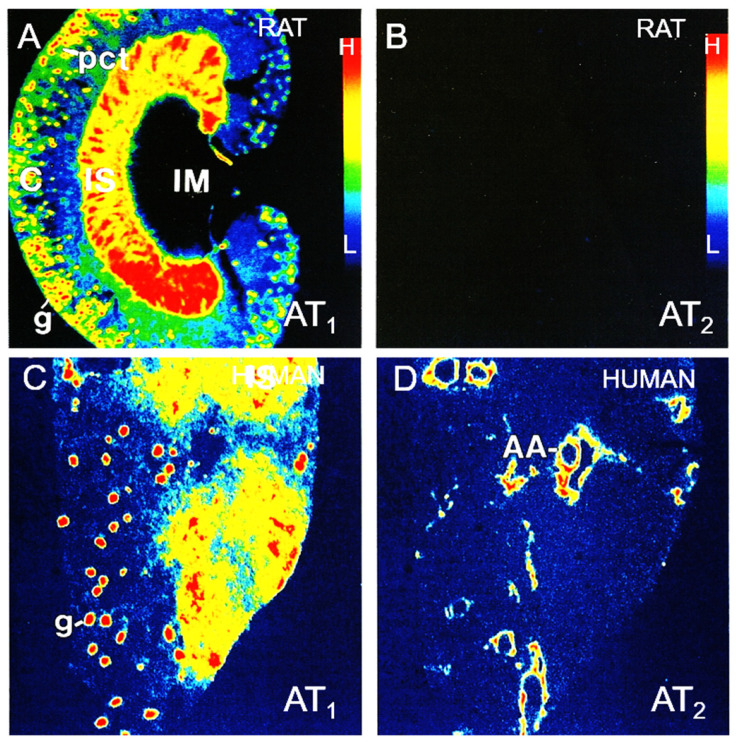
Localization of angiotensin II AT_1_ and AT_2_ receptors in the rat (**A**,**B**) and human kidney (**C**,**D**) by quantitative in vitro autoradiography using radiolabeled [^125^I]-angiotensin II as a ligand. AT_1_ receptor binding is defined as [^125^I]-angiotensin II binding in the presence of an excess concentration of unlabeled AT_2_ receptor blocker PD123319 (10 µM), whereas AT_2_ receptor binding is defined as [^125^I]-angiotensin II binding in the presence of an excess concentration of unlabeled AT_1_ receptor blocker losartan (10 µM), respectively. Red represents the highest level of binding, whereas blue represents the background level of binding. C, cortex; IS, inner stripe of the outer medulla; IM, inner medulla; AA, blood vessels including afferent and efferent arterioles and interlobular arteries; g, glomerulus; pct, proximal convoluted tubule. Modified from References [46,47,48].

**Figure 2 ijms-23-02402-f002:**
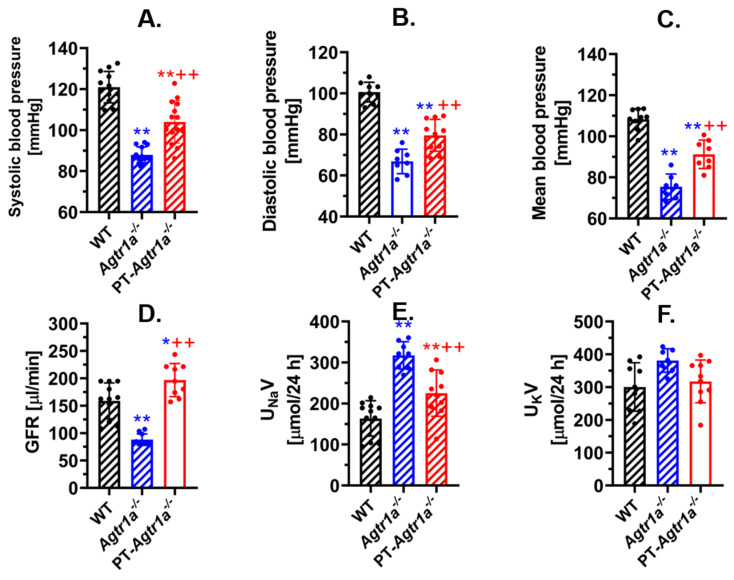
Effects of global or proximal tubule-selective deletion of AT_1a_ receptors on basal systolic, diastolic, and mean arterial blood pressure (**A**–**C**), glomerular filtration rate (**D**), and natriuretic response (**E**) in whole-body *Agtr1a^-/-^* or proximal tubule-specific PT-*Agtr1a^-/-^* mice compared with wild-type mice. Note that *Agtr1a^-/-^* mice showed that basal blood pressure was ~30 mmHg lower than wild-type mice, whereas PT-*Agtr1a^-/-^* mice had basal blood pressure ~15 mmHg lower than wild-type mice, respectively. (**F**) shows no changes in 24 h urinary potassium excretion. * *p* < 0.05 or ** *p* < 0.01 vs. WT mice; ^++^ *p* < 0.01 vs. global *Agtr1a^-/-^* mice. Reproduced from Reference [28] with permission.

**Figure 3 ijms-23-02402-f003:**
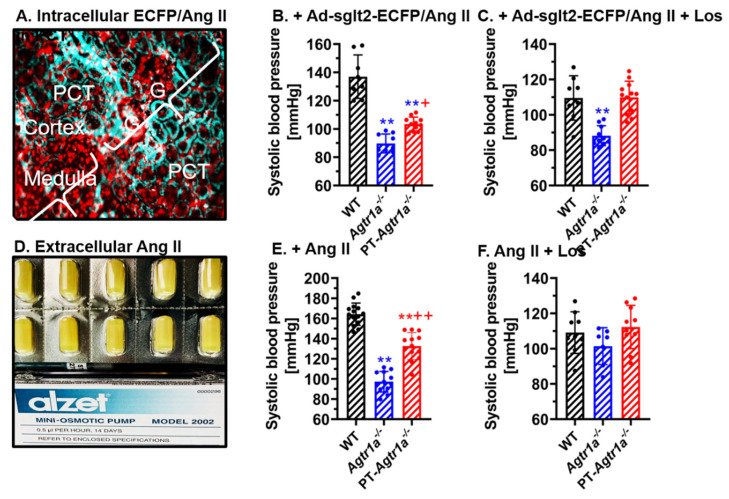
Angiotensin II-induced hypertension by adenovirus-mediated, sglt2 promoter-driven proximal tubule-specific overexpression of an intracellular Ang II fusion protein Ad-sglt2-ECFP/Ang II in the kidney (**A**–**C**), or osmotic minipump infusion of a pressor dose of Ang II (**D**–**F**) in wild-type mice was significantly attenuated in proximal tubule-specific PT-*Agtr1a^-/-^* mice. In response to proximal tubule-specific overexpression of Ad-sglt2-ECFP/Ang II, blood pressure was elevated by 17 ± 3 mmHg in wild-type mice, and the increase was totally blocked in PT-*Agtr1a^-/-^* mice (**B**). Infusion of a high pressor dose of Ang II for 2 weeks (1.5 mg/kg/day, i.p.) markedly increased blood pressure by >45 mmHg in wild-type mice, and this increase was attenuated by half in PT-*Agtr1a^-/-^* mice (**E**). Concurrent treatment with the AT_1_ receptor blocker losartan (20 mg/kg/day, p.o.) normalized blood pressure to the wild-type control levels. ** *p* < 0.01 vs. wild type; ^+^ *p* < 0.05 or ^++^ *p* < 0.01 vs. *Agtr1a^-/-^* mice. Reproduced from Reference [28] with permission.

**Figure 4 ijms-23-02402-f004:**
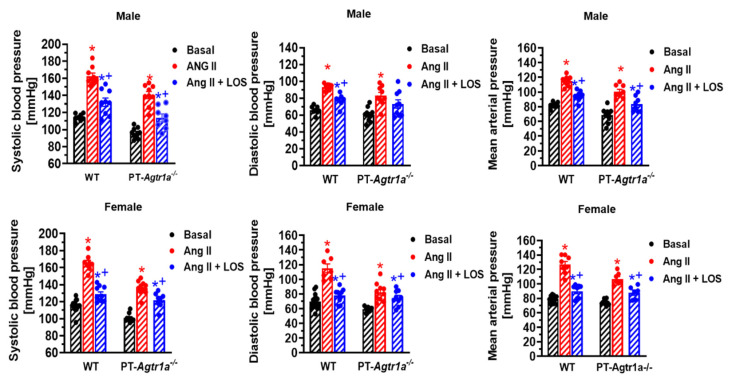
Basal systolic, diastolic, and mean arterial blood pressure and in response to osmotic minipump infusion of a high pressor dose of Ang II with or without AT_1_ (AT_1a_) receptor blocker losartan for two weeks in adult male and female wild-type and PT-*Agtr1a^-/-^* mice. Note that deletion of AT_1a_ receptors selectively in the proximal tubules significantly decreased basal blood pressure by ~13–15 ± 3 mmHg similarly in male and female PT-*Agtr1a^-/-^* mice under basal conditions. In addition, Ang II-induced hypertension was significantly attenuated in both male and female PT-*Agtr1a^-/-^* mice. * *p* < 0.05 vs. control WT or PT-*Agtr1a^-/-^* mice; ^+^ *p* < 0.05 vs. Ang II-infused male or female wild-type or PT-*Agtr1a^-/-^* mice. Reproduced from Reference [12] with permission.

## Data Availability

Not applicable.

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
