# Peer review of "Angiotensin II and AT_1a_ Receptors in the Proximal Tubules of the Kidney: New Roles in Blood Pressure Control and Hypertension"

_ijms, 2022, doi:10.3390/ijms23052402_

Round 1

Reviewer 1 Report

The manuscript „Angiotensin II and AT1a receptors in the proximal tubules of the kidney: new roles in blood pressure control and hypertension“ is a review presenting the role of Ang II in the proximal tubules in the regulation of blood pressure. The authors consider the interaction of AngII and proximal tubules to be one of the main factors in poorly controlled hypertension. They describe pathomechanisms in which Ang II / AT1a receptors / Na + / H + exchanger 3 affect water and sodium reabsorption under physiological conditions, in the development of hypertension or its resistance to treatment, and in a model of AngII-induced hypertension. The authors also present some idea of ​​gender differences in AngII-induced hypertension.

Comments:

The review is concise, straightforward and well-divided into chapters. A number of the latest findings concerning the regulatory role of the kidneys in blood pressure control are presented, including data from the own laboratory, which makes the text attractive to readers. This is a well written and interesting review.

I have only two small comments:

A list of abbreviations could help improve the readability of the text.

The abstract is too general. Instead of a relatively long introduction about well-known facts, some specific findings could be presented focusing on new data and approaches to AngII-AT1a interaction in renal tubules (such as lines 373-383).

Author Response

We would like to thank this expert reviewer for the constructive comments and suggestions on the originally submitted manuscript. We also thank the reviewer for the encouragement. Specifically, the reviewer commented that the abstract is too general, we accept the reviewer's suggestion to have revised the abstract accordingly. The reviewer also suggested to include a list of abbreviations, which is a great idea and we have included one for the effect. 

Reviewer 2 Report

Review article by Ana et al., entitled "Angiotensin II and AT1a receptor in the proximal tubules of the kidney: new roles in blood pressure control and hypertension". This is a well written review and there is no major concerns to add.  

Minor comments: 

  1. The background of the bar diagrams needs to revise with lite background. 
  2. Proximal tubule Na+ transport, and renal medullary blood flow to induce diuresis and natriuresis responses- authors need to briefly discuss the role of Natriuretic peptide receptor A (NPRA). 
  3. Similar to NPRA, discuss the role of Prorenin receptor (PRR) role in proximal tubule function. 

Author Response

Dear Reviewer,

We would like to thank you for your constructive comments and suggestions on our manuscript.

Specifically, we have considered to lighten the background of bar graphs as the reviewer suggested, but these are published data and figures reproduced from other original articles. We feel that it may be better to keep the ways in their original publications. 

As the reviewer suggested, we have briefly discussed in a new paragraph the potential roles of atrial natriuretic peptide receptor A, dopamine receptors, and prorenin receptors (PRR) in the proximal tubules involving in the blood pressure and natriuretic phenotypes in PT-Agtr1a-/- mice. However, we feel that detailed reviewing and discussing these receptors in the proximal tubules and their potential involvements in these PT-Agtr1a-/- mice are beyond the scope of this manuscript. Indeed, there is no study having been reported using these proximal tubule-specific PT-Agtr1a-/- mice to investigate the interactions between these receptors and AT1a receptors in the proximal tubules. Nevertheless, your comments and encouragement are greatly appreciated. 

In an unpublished study, we have found that deletion of AT1a receptors selectively in the proximal tubules significantly augments the natriuretic responses to ANP infusion in PT-Agtr1a-/- mice. However, these data have not been published except in a conference abstract.